# DRIFT DETECTION IN EPISODIC DATA: DETECT WHEN YOUR AGENT STARTS FALTERING

## ABSTRACT

Detection of deterioration of agent performance in dynamic environments is challenging due to the non-i.i.d nature of the observed performance. We consider an episodic framework, where the objective is to detect when an agent begins to falter. We devise a hypothesis testing procedure for non-i.i.d rewards, which is optimal under certain conditions. To apply the procedure sequentially in an online manner, we also suggest a novel Bootstrap mechanism for False Alarm Rate control (BFAR). We demonstrate our procedure in problems where the rewards are not independent, nor identically-distributed, nor normally-distributed. The statistical power of the new testing procedure is shown to outperform alternative tests – often by orders of magnitude – for a variety of environment modifications (which cause deterioration in agent performance). Our detection method is entirely external to the agent, and in particular does not require model-based learning. Furthermore, it can be applied to detect changes or drifts in any episodic signal.

## 1 INTRODUCTION

Reinforcement learning (RL) algorithms have recently demonstrated impressive success in a variety of sequential decision-making problems (Badia et al., 2020; Hessel et al., 2018). While most RL works focus on the maximization of rewards under various conditions, a key issue in real-world RL tasks is the safety and reliability of the system (Dulac-Arnold et al., 2019; Chan et al., 2020), arising in both offline and online settings.

In **offline settings**, comparing the agent performance in different environments is important for generalization (e.g., in sim-to-real and transfer learning). The comparison may indicate the difficulty of the problem or help to select the right learning algorithms. Uncertainty estimation, which could help to address this challenge, is currently considered a hard problem in RL, in particular for model-free methods (Yu et al., 2020).

In **online settings**, where a fixed, already-trained agent runs continuously, its performance may be affected (gradually or abruptly) by changes in the controlled system or its surroundings, or when reaching new states beyond the ones explored during the training. Some works address the robustness of the agent to such changes (Lecarpentier & Rachelson, 2019; Lee et al., 2020). However, noticing the changes may be equally important, as it allows us to fall back into manual control, send the agent to re-train, guide diagnosis, or even bring the agent to halt. This is particularly critical in real-world problems such as health care and autonomous driving (Zhao et al., 2019), where agents are required to be fixed and stable: interventions in the policy are often limited or forbidden (Matsushima et al., 2020), but any performance degradation should be detected as soon as possible.

Many sequential statistical tests exist for detection of mean degradation in a random process. However, common methods (Page, 1954; Lan, 1994; Harel et al., 2014) assume independent and identically distributed (i.i.d) samples, while in RL the feedback from the environment is usually both highly correlated over consecutive time-steps, and varies over the life-time of the task (Korenkevych et al., 2019). This is demonstrated in Fig. 1.

A possible solution is to apply statistical tests to large blocks of time-steps assumed to be i.i.d (Ditzler et al., 2015). Since many RL applications consist of repeating episodes, such a blocks-partition can be applied in a natural way (Colas et al., 2019). However, this approach requires complete episodes for change detection, while a faster response is often required. Furthermore, naively ap-

plying a statistical test on the accumulated feedback (e.g., sum of rewards) from complete episodes, ignores the dependencies within the episodes and may miss vital information, leading to highly sub-optimal tests.

In this work, we devise an optimal test for detection of degradation of the rewards in an episodic RL task (or in any other episodic signal), based on the covariance structure within the episodes. Even in absence of the assumptions that guarantee its optimality, the test is still asymptotically superior to the common approach of comparing the mean (Colas et al., 2019). The test can detect changes and drifts in both the offline and the online settings defined above. In addition, for the online settings, we suggest a novel Bootstrap mechanism to control the False Alarm Rate (BFAR) through adjustment of test thresholds in sequential tests of episodic signals. The suggested procedures rely on the ability to estimate the correlations within the episodes, e.g., through a "reference dataset" of episodes.

Since the test is applied directly to the rewards, it is model-free in the following senses: the underlying process is not assumed to be known, to be Markov, or to be observable at all (as opposed to other works, e.g., Banerjee et al. (2016)), and we require no knowledge about the process or the running policy. Furthermore, as the rewards are simply referred to as episodic time-series, the test can be similarly applied to detect changes in any episodic signal.

We demonstrate the new procedures in the environments of Pendulum (OpenAI), HalfCheetah and Humanoid (MuJoCo; Todorov et al., 2012). BFAR is shown to successfully control the false alarm rate. The covariance-based degradation-test detects degradation faster and more often than three alternative tests – in certain cases by orders of magnitude.

Section 3 formulates the offline setup (individual tests) and the online setup (sequential tests). Section 4 introduces the model of an episodic signal, and derives an optimal test for degradation in such a signal. Section 5 shows how to adjust the test for online settings and control the false alarm rate. Section 6 describes the experiments, Section 7 discusses related works and Section 8 summarizes.

To the best of our knowledge, we are the first to exploit the covariance between rewards in post-training phase to test for changes in RL-based systems. The contributions of this paper are (i) a new framework for model-free statistical tests on episodic (non-i.i.d) data with trusted reference-episodes; (ii) an optimal test (under certain conditions) for degradation in episodic data; and (iii) a novel bootstrap mechanism that controls the false alarm rate of sequential tests on episodic data.

## 2 PRELIMINARIES

**Reinforcement learning and episodic framework:** A Reinforcement Learning (RL) problem is usually modeled as a *decision process*, where a learning *agent* has to repeatedly make decisions that affect its future states and rewards. The process is often organized as a finite sequence of time-steps (an *episode*) that repeats multiple times in different variants, e.g., with different initial states. Common examples are board and video games (Brockman et al., 2016), as well as more realistic problems such as repeating drives in autonomous driving tasks.

Once the agent is fixed (which is the case in the scope of this work), the rewards of the decision process essentially reduce to a (decision-free) random process $\{X_t\}_{t=1}^n$, which can be defined by its PDF ($f_{\{X_t\}_{t=1}^n} : \mathbb{R}^n \to [0, \infty)$). $\{X_t\}$ usually depend on each other: even in the popular *Markov Decision Process* (Bellman, 1957), where the dependence goes only a single step back, long-term correlations may still carry information if the states are not observable by the agent.

**Hypothesis tests:** Consider a parametric probability function $p(X|\theta)$ describing a random process, and consider two different hypotheses $H_0, H_A$ determining the value (*simple hypothesis*) or allowed values (*complex hypothesis*) of $\theta$. When designing a test to decide between the hypotheses, the basic metrics for the test efficacy are its *significance* $P(\text{not reject } H_0|H_0) = 1 - \alpha$ and its *power* $P(\text{reject } H_0|H_A) = \beta$. A statistical hypothesis test with significance $1 - \alpha$ and power $\beta$ is said to be *optimal* if any test with as high significance $1 - \tilde{\alpha} \geq 1 - \alpha$ has smaller power $\tilde{\beta} \leq \beta$.

The likelihood of the hypothesis $H : \theta \in \Theta$ given data $X$ is defined as $L(H|X) = \sup_{\theta \in \Theta} p(X|\theta)$. According to Neyman-Pearson Lemma (Neyman et al., 1933), a threshold-test on the likelihood ratio $LR(H_0, H_A|X) = L(H_0|X)/L(H_A|X)$ is optimal. In a threshold-test, the threshold is uniquely determined by the desired significance level $\alpha$, though is often difficult to calculate given $\alpha$.

In many practical applications, a hypothesis test is repeatedly applied as the data change or grow, a procedure known as a *sequential test*. If the null hypothesis $H_0$ is true, and any individual hypothesis test falsely rejects $H_0$ with some probability $\alpha$, then the probability that at least one of the multiple tests will reject $H_0$ is $\alpha_0 > \alpha$, termed *family-wise type-I error* (or *false alarm rate* when associated with frequency). See Appendix K for more details about hypothesis testing and sequential tests in particular.

Common approaches for sequential tests, such as CUSUM (Page, 1954; Ryan, 2011) and $\alpha$-spending functions (Lan, 1994; Pocock, 1977), usually require strong assumptions such as independence or normality, as further discussed in Appendix F.

## 3 PROBLEM SETUP

In this work, we consider two setups where detecting performance deterioration is important – sequential degradation-tests and individual degradation-tests. The individual tests, in addition to their importance in (offline) settings such as sim-to-real and transfer learning, are used in this work as building-blocks for the (online) sequential tests.

Both setups assume a fixed agent that was previously trained, and aim to detect whenever the agent performance begins to deteriorate, e.g., due to environment changes. The ability to notice such changes is essential in many real-world problems, as explained in Section 1.

**Setup 1** (**Individual degradation-test).** We consider a fixed trained agent (policy must be fixed but is not necessarily optimal), whose rewards in an episodic environment (with episodes of length $T$) were previously recorded for multiple episodes (the *reference dataset*). The agent runs in a new environment for $n$ time-steps (both $n < T$ and $n \geq T$ are valid). The goal is to decide whether the rewards in the new environment are smaller than the original environment or not. If the new environment is identical, the probability of a false alarm must not exceed $\alpha$.

**Setup 2** (**Sequential degradation-test).** As in Setup 1, we consider a fixed trained agent with recorded reference data of multiple episodes. This time the agent keeps running in the same environment, and at a certain point in time its rewards begin to deteriorate, e.g., due to changes in the environment. The goal is to alert to the degradation as soon as possible. As long as the environment has not changed, the probability of a false alarm must not exceed $\alpha_0$ during a run of $\tilde{h}$ episodes.

Note that while in this work the setups focus on degradation, they can be easily modified to look for any change (as positive changes may also indicate the need for further training, for example).

## 4 OPTIMIZATION OF INDIVIDUAL DEGRADATION-TESTS

To tackle the problem of Setup 1, we first define the properties of an episodic signal and the general assumptions regarding its degradation.

**Definition 4.1** (*$T$-long episodic signal*). *Let $n, T \in \mathbb{N}$, and write $n = KT + \tau_0$ (for non-negative integers $K, \tau_0$ with $\tau_0 \leq T$). A sequence of real-valued random variables $\{X_t\}_{t=1}^n$ is a $T$-long episodic signal, if its joint probability density function can be written as*

$$f_{\{X_t\}_{t=1}^n}(x_1, ..., x_n) = \left[ \prod_{k=0}^{K-1} f_{\{X_t\}_{t=1}^T}(x_{kT+1}, ..., x_{kT+T}) \right] \cdot f_{\{X_t\}_{t=1}^{\tau_0}}(x_{KT+1}, ..., x_{KT+\tau_0}) \quad (1)$$

*(where an empty product is defined as 1). We further denote $\boldsymbol{\mu_0} := E[(X_1, ..., X_T)^\top] \in \mathbb{R}^T$, $\Sigma_0 := Cov((X_1, ..., X_T)^\top, (X_1, ..., X_T)) \in \mathbb{R}^{T \times T}$.*

Note that the episodic signal consists of i.i.d episodes, but is not assumed to be independent or identically-distributed within the episodes. For simplicity we focus on one-dimensional episodic signals, although a generalization to multidimensional signals is straight-forward (see Appendix G).

In the analysis below we assume that both $\boldsymbol{\mu_0}$ and $\Sigma_0$ are known. In practice, this can be achieved either through detailed domain knowledge, or by estimation from the recorded reference dataset of Setup 1, assuming it satisfies Eq. (1). The estimation errors decrease as $\mathcal{O}(1/\sqrt{N})$ with the number $N$ of reference episodes, and are distributed according to the Central Limit Theorem (for means)

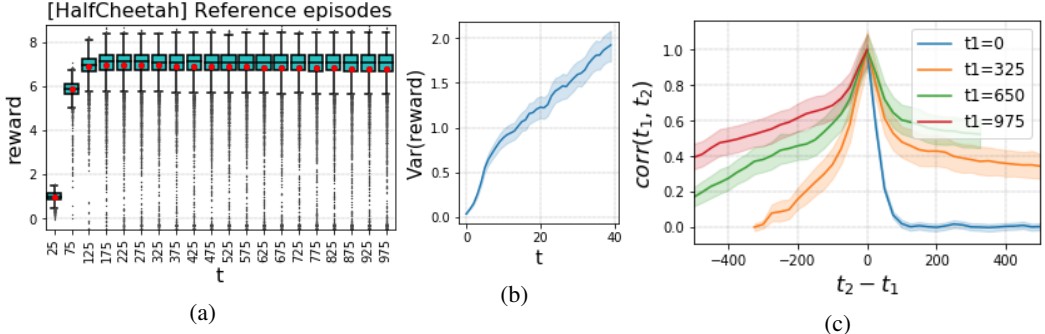

Figure 1: Parameters of an episodic signal of the rewards in HalfCheetah environment, estimated over $N = 10000$ episodes of $T = 1000$ time-steps: (a) distribution of rewards per time-step; (b) variance per time-step; (c) correlation($t_1, t_2$) vs. $t_2 - t_1$. The estimations were done in resolution of 25 time-steps, i.e., every episode was split into 40 intervals of 25 consecutive rewards, and each sample is the average over an interval.

and Wishart distribution (K. V. Mardia & Bibby, 1979) (for covariance). While in this work we use up to $N = 10000$ reference episodes, Appendix E shows that $N = 300$ reference episodes are sufficient for reasonable results in HalfCheetah, for example. Note that correlations estimation has been already discussed in several other RL works (Alt et al., 2019).

Fig. 1 demonstrates the estimation of mean and covariance parameters for a trained agent in the environment of HalfCheetah, from a reference dataset of $N = 10000$ episodes. This also demonstrates the non-trivial correlations structure in the environment. According to Fig. 1b, the variance in the rewards varies and does not seem to reach stationarity within the scope of an episode. Fig. 1c shows the autocorrelation function $ACF(t_2 - t_1) = corr(t_1, t_2)$ for different reference times $t_1$. It is evident that the correlations last for hundreds of time-steps, and depend on the time $t_1$ rather than merely on the time-difference $t_2 - t_1$. This means that the autocorrelation function is not expressive enough for the actual correlations structure.

Once the per-episode parameters $\boldsymbol{\mu_0} \in \mathbb{R}^T, \Sigma_0 \in \mathbb{R}^{T \times T}$ are known, the expectations and covariance matrix of the whole signal $\boldsymbol{\mu} \in \mathbb{R}^n, \Sigma \in \mathbb{R}^{n \times n}$ can be derived directly: $\boldsymbol{\mu}$ consists of periodic repetitions of $\boldsymbol{\mu_0}$, and $\Sigma$ consists of copies of $\Sigma_0$ as $T \times T$ blocks along its diagonal. For both parameters, the last repetition is cropped if $n$ is not an integer multiplication of $T$. In other words, by taking advantage of the episodic setup, we can treat the temporal univariate non-i.i.d signal as a multivariate signal with easily-measured mean and covariance – even if the signal is measured in the middle of an episode.

The degradation in the signal $X = \{X_t\}_{t=1}^n$ is defined through the difference between two hypotheses. The null hypothesis $H_0$ states that $X$ is a $T$-long episodic signal with expectations $\boldsymbol{\mu_0} \in R^T$ and invertible covariance matrix $\Sigma_0 \in R^{T \times T}$. Our first alternative hypothesis – uniform degradation – states that $X$ is a $T$-long episodic signal with the same covariance $\Sigma_0$ but smaller expectations: $\exists \epsilon \geq \epsilon_0, \forall 1 \leq t \leq T : (\boldsymbol{\mu})_t = (\boldsymbol{\mu_0})_t - \epsilon$. Note that this hypothesis is complex ($\epsilon \geq \epsilon_0$), where $\epsilon_0$ can be tuned according to the minimal degradation magnitude of interest. In fact, Theorem 4.1 shows that the optimal corresponding test is independent of the choice of $\epsilon_0$.

**Theorem 4.1** (Optimal test for uniform degradation). *Define the uniform-degradation weighted-mean $s_{unif}(X) := W \cdot X$, where $W := \mathbf{1}^\top \cdot \Sigma^{-1} \in \mathbb{R}^n$ (and $\mathbf{1}$ is the all-1 vector). If the distribution of $X$ is multivariate normal, then a threshold-test on $s_{unif}$ is optimal.*

*Proof Sketch.* According to Neyman-Pearson Lemma (Neyman et al., 1933), a threshold-test on the likelihood-ratio (LR) between $H_0$ and $H_A$ is optimal. Since $H_A$ is complex, the LR is a minimum over $\epsilon \in [\epsilon_0, \infty)$. Lemma 1 shows that $\exists s_0 : s_{unif} \geq s_0 \Rightarrow \epsilon = \epsilon_0$ and $s_{unif} \leq s_0 \Rightarrow \epsilon = \epsilon(s_{unif})$. The rest of the proof in Appendix J substitutes $\epsilon$ in both domains of $s_{unif}$ to prove monotony of the LR in $s_{unif}$, from which we can conclude monotony in $s_{unif}$ over all $\mathbb{R}$. $\square$

Following Theorem 4.1, we define the Uniform Degradation Test (**UDT**) to be a threshold-test on $s_{unif}$, i.e., "declare a degradation if $s_{unif} < \kappa$" for a pre-defined $\kappa$.

Recall that optimality of a test is defined in Section 2 as having maximal power given significance level. To achieve the significance $\alpha$ required in Setup 1, we apply a bootstrap mechanism that randomly samples episodes from the reference dataset and calculates the corresponding statistic (e.g., $s_{unif}$). This yields a bootstrap-estimate of the distribution of the statistic under $H_0$, and the $\alpha$-quantile of the estimated distribution is chosen as the test-threshold ($\kappa = q_\alpha(s_{unif}|H_0)$).

Note that Theorem 4.1 relies on multivariate normality assumption, which is often too strong for real-world applications. Theorem 4.2 guarantees that if we remove the normality assumption, it is still beneficial to look into the episodes instead of considering them as atomic blocks; that is, UDT is still asymptotically better than a test on the simple mean $s_{simp} = \sum_{t=1}^{n} X_t/n$. Note that "asymptotic" refers to the signal length $n \rightarrow \infty$ (while $T$ remains constant), and is translated in the sequential setup into a "very long lookback-horizon $h$" (rather than very long running time).

**Theorem 4.2** (Asymptotic power of UDT). *Denote the length of the signal $n = K \cdot T$, assume a uniform degradation of size $\frac{\epsilon}{\sqrt{K}}$, and let two threshold-tests $\tau_{simp}$ on $s_{simp}$ and UDT on $s_{unif}$ be tuned to have significance $\alpha$. Then*

$$lim_{K \rightarrow \infty} P\left(\tau_{simp} \text{ rejects } H_0 \big| H_A\right) = \Phi\left(q_\alpha^0 + \frac{\epsilon T}{\sqrt{\mathbf{1}^\top \Sigma_0 \mathbf{1}}}\right)$$

$$\leq \Phi\left(q_\alpha^0 + \epsilon\sqrt{\mathbf{1}^\top \Sigma_0^{-1} \mathbf{1}}\right) = lim_{K \rightarrow \infty} P\left(\text{UDT rejects } H_0 \big| H_A\right)$$

(2)

*where $\Phi$ is the CDF of the standard normal distribution, and $q_\alpha^0$ is its $\alpha$-quantile.*

*Proof Sketch.* Since the episodes of the signal are i.i.d, both $s_{simp}$ and $s_{unif}$ are asymptotically normal according to the Central Limit Theorem. The means and variances of both statistics are calculated in Lemma 2. Calculation of the variance of $s_{unif}$ relies on writing $s_{unif}$ as a sum of linear transformations of $X$ ($s_{unif} = \sum_{i=1}^{n} (\Sigma^{-1})_i X$), and using the relation between $\Sigma$ and $\Sigma_0$. Appendix J shows that the inequality between the resulted powers is equivalent to a matrix-form of the means-inequality, and proves it by applying Cauchy-Schwarz inequality to $\Sigma_0^{-1/2} \mathbf{1}$ and $\Sigma_0^{1/2} \mathbf{1}$. $\square$

Motivated by Theorem 4.2, we define $G^2 := \frac{(\mathbf{1}^\top \Sigma_0^{-1} \mathbf{1})(\mathbf{1}^\top \Sigma_0 \mathbf{1})}{T^2}$ to be the asymptotic power gain of UDT, quantify it, and show that it increases with the heterogeneity of the spectrum of $\Sigma_0$.

**Proposition 4.1** (Asymptotic power gain). *$G^2 = 1 + \sum_{i,j=1}^{T} w_{ij}(\lambda_i - \lambda_j)^2$, where $\{\lambda_i\}_{i=1}^{T}$ are the eigenvalues of $\Sigma_0$ and $\{w_{ij}\}_{i,j=1}^{T}$ are positive weights.*

*Proof Sketch.* The result can be calculated after diagonalization of $\Sigma_0$, and the weights $\{w_{ij}\}$ correspond to the diagonalizing matrix. See Appendix J for more details. $\square$

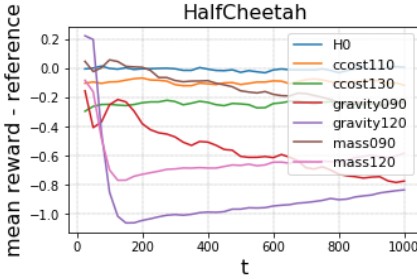

Figure 2: Rewards degradation of a fixed agent in HalfCheetah following changes in gravity, mass, and control-cost, over $N = 5000$ episodes per scenario.

So far we assumed a uniform degradation. In the context of RL, such a model may refer to changes in constant costs or action costs, as well as certain environment dynamics whose change influences various states in a similar way. Fig. 2 demonstrates the empiric degradation in the rewards of a fixed agent in HalfCheetah, following changes in gravity, mass and control-cost. It seems that some modifications indeed cause a quite uniform degradation, while in others the degradation is mostly restricted to certain ranges of time.

To model effects that are less uniform in time we suggest a partial degradation hypothesis, where some (unknown) entries of $\boldsymbol{\mu_0}$ are reduced by $\epsilon > 0$, and others do not change. The number $m = p \cdot T$ of the reduced entries is defined by a parameter $p \in (0, 1)$.

This time, calculation of the optimal test-statistic through the LR yields a minimum over $\binom{T}{m}$ possible subsets of decreased entries, which is computationally heavy. However, Theorem 4.3 shows that if we optimize for small values of $\epsilon$ (where optimality is indeed most valuable), a near-optimal statistic is $s_{part}$, which is the sum of the $m = p \cdot T$ smallest time-steps of $(X - \boldsymbol{\mu})$ after a $\Sigma_0^{-1}$-transformation (see formal definition in Definition I.11). We define the Partial Degradation Test (**PDT**) to be a threshold-test on $s_{part}$ with a parameter $p$.

**Theorem 4.3** (Near-optimal test for uniform degradation). *Assume that $X$ is multivariate normal, and let $P_\alpha$ be the maximal power of a hypothesis test with significance $1 - \alpha$. The power of a threshold-test on $s_{part}$ with significance $1 - \alpha$ is $P_\alpha - \mathcal{O}(\epsilon)$.*

*Proof Sketch.* The expression that is minimized is a sum of two terms. One term is the sum of a subset of entries of $\Sigma^{-1}(X - \boldsymbol{\mu})$, which is minimized by simply taking the lowest entries (up to the constraint of consistency across episodes, which requires us to sum the rewards per time-step in advance). In Appendix J we bound the second term and its effects on the modified statistic and on the modified test-threshold. We show that the resulted decrease of rejection probability is $\mathcal{O}(\epsilon)$. □

## 5 BOOTSTRAP FOR FALSE ALARM RATE CONTROL (BFAR)

For Setup 2, we suggest a sequential testing procedure: run an individual degradation-test every $d$ steps (i.e., $F = T/d$ test-points per episode), and return once any individual test declares a degradation. The tests can run according to Section 4, applied on the $h$ recent episodes. Multiple tests may be applied every test-point, e.g., with varying test-statistics $\{s\}$ or lookback-horizons $\{h\}$. This procedure, as implemented for the experiments of Section 6, is described in Fig. 3.

Setup 2 limits the probability of a false alarm to $\alpha_0$ in a run of $\tilde{h}$ episodes. To satisfy this condition, we set a uniform threshold $\kappa$ on the $p$-values of the individual tests (i.e., declare once a test returns $p$-val $< \kappa$). The threshold is determined using a Bootstrap mechanism for False Alarm control (**BFAR**, Algorithm 1).

While bootstrap methods for false alarm control are quite popular, they often rely on the data samples being i.i.d (Kharitonov et al., 2015; Abhishek & Mannor, 2017), which is crucial for the re-sampling to reliably mimic the source of the signal. To address the non-i.i.d signal, we take advantage of the episodic framework and sample whole episodes. We then use the re-sampled sequence to simulate tests on sub-sequences where the first and last episodes may be incomplete, as described below. This allows simulation of sequences of various lengths (including non-integer number of episodes) without assuming independence, normality, or identical distributions within the episodes.

---

**Algorithm 1:** BFAR: Bootstrap for FAR control

---

**Input**: reference dataset $x \in \mathbb{R}^{N \times T}$; statistic functions $\{s\}$; lookback-horizons $\{h_1, ..., h_{max}\}$; test length $\tilde{h} \in \mathbb{N}$; $B \in \mathbb{N}$; $\alpha_0 \in (0, 1)$;
**Output**: test threshold for individual tests;
Initialize $P = (1, ..., 1) \in [0, 1]^B$;
**for** *b in 1:B* **do**
    Initialize $Y \in \mathbb{R}^{(h_{max} + \tilde{h})T}$;
    **for** *k in 0:($h_{max}$+$\tilde{h}$-1)* **do**
        Sample $j$ uniformly from $(1, ..., N)$;
        $Y[kT + 1 : kT + T] \leftarrow (x_{j1}, ..., x_{jT})$;
    **for** *t in test-points* **do**
        **for** *h in lookback-horizons and s in statistic functions* **do**
            $y \leftarrow Y[t - hT : t]$;
            $p \leftarrow$ individual_test_pvalue($y$ vs. $x$; $s$);
            $P[b] \leftarrow \min(P[b], p)$;
Return $quantile_{\alpha_0}(P)$;

---

BFAR samples $h_{max} + \tilde{h}$ episodes (where $h_{max}$ is the maximal lookback-horizon) from reference data of $N$ episodes, to simulate sequential data $Y$. Then individual tests are simulated for any test-point along $\tilde{h}$ episodes, starting after $h_{max}$ episodes. The minimal $p$-value determines whether a detection would occur in $Y$. The whole procedure repeats $B$ times, creating a bootstrap estimate of the distribution of the minimal $p$-value along $\tilde{h}$ episodes. We choose the tests threshold to be the $\alpha_0$-quantile of this distribution, such that $\alpha_0$ of the bootstrap simulations would raise a false alarm.

Note that the statistic for the tests is given to BFAR as an input, making its choice independent of BFAR. Additional details and time complexity are discussed in Appendices H,L.

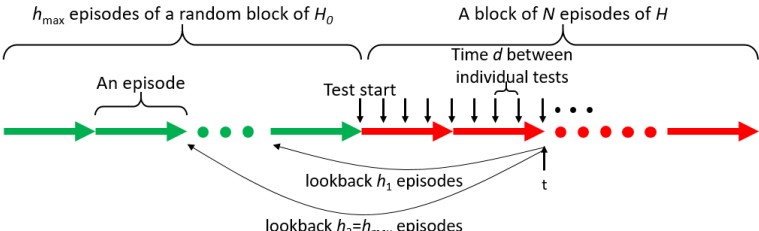

Figure 3: A summary of the sequential degradation-test procedure described in Section 6.1.

# 6 EXPERIMENTS

## 6.1 METHODOLOGY

We run experiments in standard Reinforcement Learning environments as described below. For every environment, we use a PyTorch implementation (Kostrikov, 2018) of the standard A2C algorithm (Mnih et al., 2016) to train an agent. We let the trained agent run in the environment for $N_0$ episodes and record its rewards, considered the *trusted reference data*. We then define several scenarios, and let the agent run for $M \times N$ episodes in each scenario (divided later into $M = 100$ blocks of $N$ episodes). One scenario is named $H_0$ and is identical to the reference run (up to initial-state randomization). The other scenarios are defined per environment, and present environmental changes expected to harm the agent's rewards. The agent is *not* trained to adapt to these changes, and the goal is to test how long it takes for a degradation-test to detect its degradation.

Individual degradation-tests of length $n$ (Setup 1) are applied for every scenario over the first $n$ time-steps of each block. Sequential degradation-tests (Setup 2) are applied sequentially on the episodes of each block. Since the agent is assumed to run continuously as the environment changes from $H_0$ to an alternative scenario, each block is preceded by a random sample of $H_0$ episodes, as demonstrated in Fig. 3.

BFAR adjusts the tests thresholds to have a false alarm with probability $\alpha_0 = 5\%$ per $\tilde{h} = N$ episodes (where $N$ is the data-block size). Two lookback-horizons $h_1, h_2$ are chosen for every environment. The rewards are downsampled by factor $d$ before applying the tests, intended to reduce the parameters estimation error and the running time of the tests. Table 1 summarizes the setup of the various environments.

The experimented degradation-tests are a threshold-test on the simple **Mean**; **CUSUM** (Ryan, 2011); **Hotelling** (Hotelling, 1931); **UDT** and **PDT** (with $p = 0.9$) from Section 4; and a Mixed Degradation Test (**MDT**) that runs Mean, Hotelling and PDT in parallel – applying all three in every test-point (as permitted in Algorithm 1). Further implementation details are discussed in Appendix D.

## 6.2 RESULTS

We run the tests in the environments of Pendulum (OpenAI), where the goal is to keep a one-dimensional pendulum pointing upwards; HalfCheetah (Todorov et al., 2012), where the goal is for

Table 1: Environments parameters
(episode length ($T$), reference episodes ($N_0$), test blocks ($M$), episodes per block ($N$),
sequential test length ($\tilde{h}$), lookback horizons ($h_1, h_2$), tests per episode ($F = T/d$))

| Environment | $T$ | $N_0$ | $M$ | $N(= \tilde{h})$ | $h_1, h_2$ | $F = T/d$ |
|---|---|---|---|---|---|---|
| Pendulum-v0 | 200 | 3000 | 100 | 30 | 3,30 | 20 |
| HalfCheetah-v3 | 1000 | 10000 | 100 | 50 | 5,50 | 40 |
| Humanoid | 200 | 5000 | 100 | 30 | 3,30 | 10 |

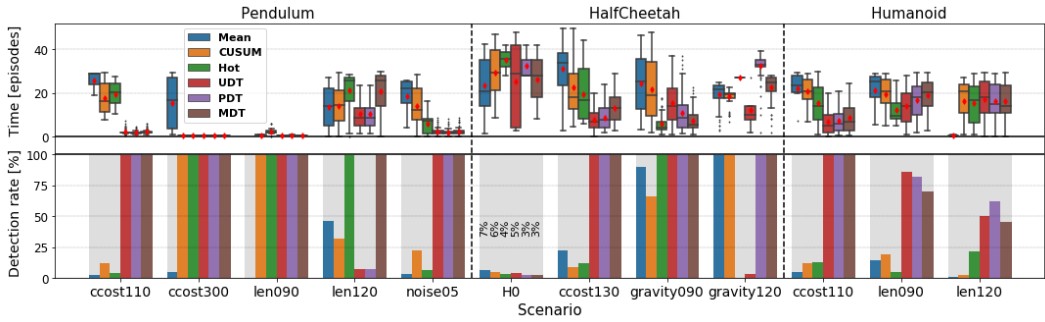

Figure 4: Bottom: percent of sequential tests (among $M = 100$ runs with different seeds) that ended with degradation detection, for various degradation-tests (corresponding to different colors), in a sample of scenarios in Pendulum, HalfCheetah and Humanoid. Top: the distribution of time until detection – for the runs that ended with detection. High detection rates usually go along with short detection times.

a two-dimensional cheetah to run as fast as possible; and Humanoid, where the goal is for a person to walk without falling. In each environment we define the scenario *ccostx* of control cost increased to x% of its original value, in addition to scenarios of changed dynamics as specified in Appendix D.

In all the environments the rewards are clearly *not* independent, identically-distributed or normally-distributed (see Fig. 1 for example). Yet the false alarm rates are close to $\alpha_0 = 5\%$ per $\tilde{h}$ episodes in all the tests, as demonstrated in Fig. 4 for HalfCheetah, for example. These results for the $H_0$ scenarios indicate that BFAR tunes the thresholds properly in spite of the complexity of the data. Note that BFAR never observed the data of scenario $H_0$, but only the reference data.

In most of the non-$H_0$ scenarios, our tests prove to be more powerful than the standard tests, often by extreme margins. For example, increased control cost in all the environments and additive noise in Pendulum are all 100%-detected by the suggested tests, usually within few episodes (Fig. 4); whereas Mean, CUSUM and Hotelling have very poor detection rates. Mean did not detect degradation in Pendulum even after the control cost increased from 110% to 300%(!).

Note that we run the tests with two lookback-horizons in parallel, as allowed by BFAR. This proves useful: with +30% control cost in HalfCheetah, for example, the short lookback-horizon allows fast detection of degradation; but with merely +10%, the long horizon is necessary to notice the slight degradation over a large number of episodes. This is demonstrated in Fig. 11 in Appendix C.

The covariance-based tests reduce the weights of the highly-varying (and presumably noisier) time-steps. In HalfCheetah they turn out to be in the later parts of the episode. As a result, in certain scenarios, Mean (which ignores the different variances), CUSUM and Hotelling (which exploit them only in a heuristic way) do better in individual degradation-tests of 100 samples (out of $T = 1000$) than they do in one or even 10 full episodes. This does not occur in UDT and PDT. Essentially, we see that ignoring the noise variability leads to violation of the principle that more data are better.

In Pendulum the ratio between variance of different steps may reach 5 orders of magnitude. This phenomenon increases the potential power of the covariance-based tests. For example, when the pole is shortened, negative changes in the highly-weighted time-steps are detected even when the mean of the whole signal increases. This feature allows us to detect slight changes in the environment before they develop into larger changes and cause damage.

On the other hand, a challenging situation arises when certain rewards decrease but the highly-weighted ones slightly increase (as in longer Pendulum's pole), which strongly violates the assumptions of Section 4. UDT is doomed to falter in such scenarios. PDT proves somewhat robust to this phenomenon since it is capable of focusing on a subset of time-steps, as demonstrated in increased gravity in HalfCheetah (see Fig. 4). However, it cannot overcome the extreme weights differences in Pendulum. The one test that demonstrated robustness to all the experimented scenarios, including modified Pendulum's length and mass, is MDT. MDT combines Mean, Hotelling and PDT and does not fall far behind any of the three, in any of the scenarios. Hence, it presents excellent results in some scenarios and reasonable results in the others.

Detailed experiments results are available in Appendix C.

## 7 RELATED WORK

Training in non-stationary environments has been widely researched, in particular in the frameworks of MAB (Mukherjee & Maillard, 2019; Garivier & Moulines, 2011; Besbes et al., 2014; Lykouris et al., 2020; Alatur et al., 2020; Gupta et al., 2019; Jun et al., 2018), model-based RL (Lecarpentier & Rachelson, 2019; Lee et al., 2020) and general multi-agent environments (Hernandez-Leal et al., 2019). Banerjee et al. (2016) explicitly detect changes in the environment and modify the policy accordingly, but assume that the environment is Markov, fully-observable, and its transition model is known – three assumptions that we avoid and that do not hold in many real-world problems. Safe exploration during training in RL was addressed by Garcia & Fernandez (2015); Chow et al. (2018); Junges et al. (2016); Cheng et al. (2019); Alshiekh (2017). Note that our work refers to changes beyond the scope of the training phase: it addresses the stage where the agent is fixed and required not to train further, in particular not in an online manner. Robust algorithms may prevent degradation in the first place, but when they fail – or when their assumptions are not met – a model-free monitor with minimal assumptions (as the one suggested in this work) is crucial.

Sequential tests were addressed by many over the years. Common approaches rely on strong assumptions such as samples independence (Page, 1954; Ryan, 2011) and normality (Pocock, 1977; O'Brien & Fleming, 1979). Generalizations exist for certain private cases (Lu & Jr., 2001; Xie & Siegmund, 2011), sometimes at cost of alternative assumptions such as known change-size (Lund et al., 2007). Samples independence is usually assumed also in recent works based on numeric approaches (Kharitonov et al., 2015; Abhishek & Mannor, 2017; Harel et al., 2014), and is often justified by consolidating many data samples (e.g., an episode) together as a single sample (Colas et al., 2019). Ditzler et al. (2015) wrote that "change detection is typically carried out by inspecting i.i.d features extracted from the incoming data stream, e.g., the sample mean". Certain works address monitoring of cyclic signals (Zhou et al., 2005), but to the best of our knowledge, we are the first to devise an optimal test for mean change in temporal non-i.i.d signals, and bootstrap-based false alarm control for such non-i.i.d signals.

Our work can be seen in part as converting a univariate temporal episodic signal into a $T$-dimensional multivariate signal (with incomplete observations in mid-episodes). Many works addressed the problem of changepoint detection in multivariate variables, e.g., using histograms comparison (Boracchi et al., 2018), Hotelling statistic (Hotelling, 1931), and K-L distance (Kuncheva, 2013). Hotelling in particular looks for changed mean under unchanged covariance, similarly to our work. However, it is not derived optimally for mean change detection, and it also inherently ignores the sign of change. Our test is optimal under similar conditions to Hotelling test, is further proved to be robust to the normality assumption, and is shown to perform better in a variety of experiments. We are not aware of any other work that derives an optimal test to either the uniform degradation or the partial degradation complex hypotheses.

## 8 SUMMARY

We introduce a novel approach that is optimal (under certain conditions) for detection of changes in episodic signals, exploiting the correlations structure as measured in a reference dataset. In environments of classic control (Pendulum) and MuJoCo (HalfCheetah, Humanoid), the suggested statistical tests detected degradation faster than alternatives, often by orders of magnitude. Certain conditions, such as combination of positive and negative changes in very heterogeneous signals, may cause instability in some of the suggested tests; however, this is shown to be solved by running the new test in parallel to standard tests – with only a small loss of test power.

We also introduce BFAR, a bootstrap mechanism that adjusts tests thresholds according to the desired false alarm rate in sequential tests. The mechanism empirically succeeded in providing valid thresholds for various tests in all the environments, in spite of the non-i.i.d data.

The suggested approach may contribute to development of more reliable RL-based systems. Future research may: consider different hypotheses, such as a permitted small degradation (instead of $H_0$) or a mix of degradation and improvement (instead of $H_A$); suggest additional stabilizing mechanisms for covariance-based tests; exploit other metrics than rewards for tests on model-based RL systems; and apply comparative tests of episodic signals beyond the scope of drifts detection.

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
