# OpenReview forum: "Drift Detection in Episodic Data: Detect When Your Agent Starts Faltering"
_ICLR.cc/2021/Conference — Reject_

### Official Review · AnonReviewer4 · 2020-10-19
**Comments to "Drift Detection in Episodic Data"**

**Rating:** 5
**Confidence:** 4

**Review:**

This paper considers the drift detection for episodic data, where data episodes are assumed to be i.i.d. but data within each episodic can be correlated. It is assumed that the pre-change (nominal) mean and covariance of each episodic is perfectly known or can be accurately estimated from reference data. The Uniform Degradation Test (UDT) and Partial Degradation Test (PDT) are proposed to detect the mean shift. Moreover, this paper uses bootstrap to control the false alarm rate by setting the threshold as empirical quantiles of the detection statistic computed from reference data.

Overall, this paper is well written and well-structured. The most impressive part is that they include comprehensive details, covering almost every related aspect. And this paper has tried to make the problem as general as possible, such as considering non-Gaussian distributions, etc.

However, the technical contribution is very incremental. And the problem itself, although is proposed in the reinforcement learning setting, is of no fundamental difference with the classical mean shift in change-point detection literature, for example, change-point detection for a mean shift in multivariate Gaussian distributions. And I didn't see the classical Hotelling T^2 test mentioned in this paper, which is a classical and also widely-used method to detect the change in mean/covariance, and it also utilizes the pre-change covariance matrix in the detection statistic. Moreover, the proposed Uniform Degradation Test methods are a direct consequence of the likelihood ratio test; the false alarm control by bootstrap is also widely used in change-point detection literature for problems where the theoretical characterization of false alarm rate (or average run length) is difficult to obtain.

Looking forward, I think this paper may be improved by expanding the horizon of the problem set-up and possibly leading to new theoretical findings.

================ After the author response, I raise my score by 1 ================
Thanks to the authors for the detailed response and extended discussion on theoretical results and experiments, and I have raised my score by 1.

---

> ### Author Response · Authors · 2020-11-13
> **We clarified the added value of our test (which is optimal), added Hotelling to the experiments, and extended the theoretical results**
>
> We thank the reviewer for the insightful comments.
>
> Hotelling test is indeed an alternative to ours, and we added it both to the discussion in Section 7 and to the experiments in Section 6. While being important and useful, this test differs from ours and is *not optimal* for the problem as defined in the paper. Indeed, as shown in the updated results (on the same scenarios as before), our statistic seems to do better than Hotelling.
>
> Regarding our contribution wrt change-point detection in general, we believe that we introduce several important contributions as described below, and we added several corresponding clarifications in sections 1,4,5,7, including additional results (that were previously mentioned only by title) in Section 4. Also see the description below for more details.
>
> We would like to know the reviewer's opinion regarding the modifications.
>
> ________________________
> Changepoint detection is a broad domain with many approaches relevant for different frameworks, some of them we referred to, and certain others we intend to add (e.g. Kuncheva, Ludmila I., 2011). Many methods are not derived as optimal wrt concrete hypotheses. We suggest a specific model that in part looks at a univariate temporal signal as multivariate. We form two different hypotheses that we believe that reflect the nature of the relevant practical problem in RL. We use LR on two different hypotheses to derive a specific test which is very simple to use and yet optimal. Each of the two hypotheses is complex wrt a different parameter space, which complicates the proof as now explained in Section 4. We also prove under minimal assumptions (no normality) our advantage over the common approach of taking data blocks as single points (see for example (Colas et al., 2019), (Ditzler et al., 2015)), and quantify this advantage in terms of Sigma's spectrum. We made efforts to clarify all these points in the updated revision.
> We could not find a straightforward optimal statistical test that handles these issues, and we would be grateful if the reviewer could refer us to relevant works.
>
> The suggested bootstrap mechanism is unique in that it can handle non-iid samples. One may say that the innovation is in representing the univariate temporal episodic signal as multivariate, rather than in the algorithm itself. However, (1) the temporal dimension does have meaning when testing in the middle of an episode; (2) in either case we are not aware to such application of bootstrap to non-iid univariate signals in general and rewards in particular. These two points are now discussed in sections 5,7.
>
> Our work combines several contributions regarding how we look at the problem (focus on rewards that require minimal knowledge & modeling, converting temporal univariate into multivariate, defining two alternative hypotheses); theoretical results (as mentioned above); and practical results (major improvement over common practices, robust way to incorporate with existing methods, bootstrap of non-iid temporal signal). While part of these contributions are arguably "incremental", we believe that as a whole we introduce a novel, important and nontrivial approach to the problem of performance degradation in episodic framework.

---

### Official Review · AnonReviewer3 · 2020-10-28
**Interesting work, it is not clear to me how much it is novel w.r.t. standard multivariate CDT.**

**Rating:** 6
**Confidence:** 3

**Review:**

The authors present a novel change detection test for non i.i.d. data motivated by applications in RL. At first, they provide an offline version of the test, then they extend it to the online setting.

The paper is clearly written and presents both theoretical results and convincing experimental results. My two concerns are about the novelty of what has been proposed w.r.t. standard CDT procedures and on the fact that a consistent part of the material of what has been proposed in the paper is deferred to the appendix.

I would like to have more discussion on the difference between what has been proposed here and the standard multivariate CDTs, for instance:
Kuncheva, Ludmila I. "Change detection in streaming multivariate data using likelihood detectors." IEEE transactions on knowledge and data engineering 25.5 (2011): 1175-1180.
Boracchi, Giacomo, et al. "Quanttree: histograms for change detection in multivariate data streams." International Conference on Machine Learning. 2018.
A strong motivation of the novelty w.r.t. to the literature might make me increaese the paper score.

In my opinion, the paper is not self-contained. Not even the main theorem proof are included (the sketches are not useful in understanding the proof line) and the experimental setting is described in details only in the additional material. I think you should rearrange some of the material from the appendix to the main paper and viceversa.

I would have appreciated a more detailed description of Algorithm 1. In this version of the paper it is difficult to understand the procedure you proposed, if you do not refer to the appendix for details.

In your setting the change in the episodic reward is only about the expected values. What happens if the new reward distribution changes in terms of covariance \Sigma?

Minor:
"in RL ... life-time of the task." I would have preferred a citation about this statement. Showing evidence using your experiment is a bit premature at this stage of the presentation.
assume strong assumptions -> require strong assumptions

--------------------------------------------------------------------------------------------------
After rebuttals the authors significantly improved the presented work, including and discussing some relevant work which was previously missing.

---

> ### Author Response · Authors · 2020-11-13
> **We extended the discussion in our innovation (e.g. optimality), and extended the experiments and the theoretical section**
>
> We thank the reviewer for the helpful comments.
>
> The suggested references are indeed relevant and were added to Section 7, along with further discussion regarding the uniqueness of our method. We also extended the description of the theoretical results and their derivation in Section 4. For more details please also see the (*) below.
> In addition, Hotelling test (which is discussed in the IEEE reference and also takes advantage of knowing Sigma) was added to the experiments in Section 6 and to the discussion in Section 7.
>
> Regarding the dependence on appendices, we made efforts to clarify the proof sketches, BFAR's description and the experimental settings. Furthermore, we extended Section 4 to include additional theoretical results. We are also open to concrete suggestions regarding reduction of specific parts that are currently in the front paper.
>
> Regarding sigma, our model assumes that it remains constant. As shown in the experiments, our test practically performs quite well even if our model does not accurately describe the process. Of course, different degradation hypotheses with modified sigma could also be considered, either independently or merged with our suggested test (using MDT for the merge).
>
> Regarding the last reviewer's comment, an external citation was added.
>
> We would like to know the reviewer's opinion regarding the modifications.
>
> _____________________
> (*) Change-point detection in general is a broad domain with many approaches relevant for different frameworks, some of them we indeed referred to. Many methods are not derived as optimal wrt concrete hypotheses. Our first key insight is that the univariate temporal episodic signal can be (partially) viewed as a multivariate variable. We then suggest a specific model & hypotheses that we believe that reflect the nature of the relevant practical problem, and we derive a specific test which is very simple to use and yet is *optimal*. This derivation handles two different complex hypotheses with different domains of parameters, each arises its own difficulties (Theorems 4.1,4.3), and also handles a possibly-unfinished last episode. We could not find a straightforward optimal statistical test that handles these issues, and we would most appreciate it if the reviewer could refer us to relevant literature.
> In addition, since normality can hardly be assumed in practical problems, we proved asymptotic advantage over the standard mean test, independently of the underlying distribution, and quantified the advantage in terms of the spectrum of Sigma (Theorem 4.2, Proposition 4.1).

---

### Official Review · AnonReviewer1 · 2020-10-28
**Seemingly incremental paper with results that may have an extended application domain**

**Rating:** 6
**Confidence:** 3

**Review:**

The authors propose a sequential testing procedure for a fixed RL agent reward drift.

Major issues:

1. The authors state in some places of the text that their method can be applied in statistic domain other than RL. However, for me, the entire proposed methodology does not relate to RL and is more general: it can be applied to any sequential statistics (series of random variables) that have a proper [auto]correlation structure (just see how the main statements). Due to this, I have concerns in the way how these proposed approaches are presented: I believe that a paper requires a notable refactoring to improve the presentation.


2. Seem that the contribution is not enough for the current venue.  All the proposed statements (Theorem 4.1, Theorem 4.2) looks very straightforward. Possibly, there untrivial tricks in their proofs, but the current proofs (proof sketch) does not reflect them at all (see). I expect a clearer presentation in the main text why these results are non-trivial and non-incremental. As for now, the provided theoretical grounds looks as non-enough for publication @ ICLR.



Other issues:
Paper cites the paper
 [Vineet Abhishek and Shie Mannor. A nonparametric sequential test for online randomized experiments. Proceedings of the 26th International Conference on World Wide Web Companion, pp.
610–6, 2017]
where seq. testing is applied to A/B tests, but there is an earlier paper on this approach:
 [Eugene Kharitonov, Aleksandr Vorobev, Craig Macdonald, Pavel Serdyukov, and Iadh Ounis. Sequential testing for early stopping of online experiments. In Proceedings of the 38th International ACM SIGIR Conference on Research and Development in Information Retrieval, SIGIR’15, pages 473–482, New York, NY, USA, 2015. ACM.]

================
After the author response, I raise my score by 1 (see my comment to them)

---

> ### Author Response · Authors · 2020-11-13
> **We extended the theoretical work described in Section 4**
>
> We thank the reviewer for his/her comments.
>
> 1. While the mathematical framework is indeed formulated in a general way, we focus on RL for two reasons: (a) episodic setup is standard in RL, and not so in most domains (even domains with cyclic signals are often not separated to independent episodes); (b) we believe that the method may facilitate getting RL to work in real-world applications with  safety requirements.
>
> 2. We take responsibility for not clarifying the full meaning of the results and the challenges in achieving them, and we made efforts to fix it in the updated revision, mainly in Section 4 (but also 5 and 7). The main details are also available in the (*) below.
>
> 3. The suggested earlier reference is indeed relevant and was added.
>
> We would like to know the reviewer's opinion regarding the modifications.
>
> _____________________
> (*) We believe that a notable innovation in our approach is the viewing of a temporal univariate episodic signal as multivariate, and the formulation of concrete alternative hypotheses from which an *optimal* test can be derived.
> The derivation itself handles the challenges of complex hypotheses with 2 different domains of parameters (which complicate the proof as now explained in theorems 4.1,4.3), as well as a possibly unfinished last episode (which corresponds to incomplete observation in the analogy of multivariate signals).
> In addition, in absence of normality, Theorem 4.2 required formulation of the conditions in a way that expresses the asymptotic advantage, and some nontrivial work in the proof of the inequality itself, as well as in the quantification of the test's advantage in terms of the spectrum of Sigma (see the added Proposition 4.1).

---

> > ### Comment · AnonReviewer1 · 2020-11-20
> > **Proof sketches are extended**
> >
> > I thank the authors for their efforts and improvement of the paper.
> >
> > For now I see that the proofs (proof sketches) are more complete and they can be used to understand the key ideas behind the proofs.
> > On the other hand,
> > (a) The proofs seem be based on standard techniques (also mentioned by Reviewer AnonReviewer4 in some sense).
> > (b) I still see the issues related to the  presentation of the results w.r.t. RL (also mentioned by Reviewer AnonReviewer3). So,  even if you have new contributions in your settings, it is still unclear why these contributions are not useful outside RL domain.
> >
> > The paper still looks as borderline, but I'm ready to up my score by 1.

---

> > > ### Author Response · Authors · 2020-11-21
> > > **We thank the reviewer and accept the comments**
> > >
> > > We thank the reviewer for the additional response and reconsideration of our work.
> > >
> > > Please note that our contribution includes a non-standard way to look at the problem of agent performance degradation, along with both theoretical and empirical improvements over alternative methods.
> > > Naturally, certain parts of our theoretical proofs indeed apply standard mathematical techniques. This does not stand in contradiction to our belief that as a whole we introduce a novel, important and nontrivial approach to the problem of performance degradation in episodic framework.
> > > As explained before, this is also one of the motivations to presenting our work from RL point of view, even though certain contributions might indeed be applied to other fields as well.

---

### Official Review · AnonReviewer2 · 2020-10-29
**Interesting and solid idea, but lacks demonstration of its use**

**Rating:** 5
**Confidence:** 3

**Review:**

This paper designed a hypothesis testing procedure for detecting changes in episode sequential data.  For online operation, it also proposed a novel Bootstrap mechanism for False alarm rate control. The method is demonstrated based on a non-iid and non-Gaussian setting for reward signals.

In all the method is theoretically sound and seems useful for detecting changes, however there is no experiments provided to show that the method can help improving the performance in RL tasks. The strength of this paper would be significantly boosted if the proposed method can be used to solve an non-stationary RL problem.

Moreover, it is a little unclear the specific setting of this work. Is the underlying sequential decision making problem based on Markov decision processes(MDPs)?
There is no surprise that the RL feedback from an environment is highly correlated over consecutive time-step if the underlying problem is an MDP. It seems that the paper mainly deal with per-step reward without considering state and action information when detecting changes.
Some related work that also applies hypothesis testing procedure for non-stationary MDPs might worth mentioning. E.g,

Banerjee et al. Quickest change detection approach to optimal control in Markov decision processes with model changes

“In setup 2, … we consider a fixed trained agent … , “ It is unclear if the trained agent exhibit an optimal behavior or arbitrary behavior(sub-optimal)?

In figure 2, does each curve correspond to a fixed policy?

=========after rebuttal===========

I appreciate the authors’ effort to address my questions. I still think this paper is below my expectation especially if it is put into the context of RL.  I would expect to see how this method can solve or help solving a fundamental problem (e.g., reducing sample complexity) in RL or be applied to a novel application (e.g., nonstationary RL tasks). Otherwise I didn’t see why it is necessary to compare this method with other statistical testing methods in the context of RL, if they can be easily made in other non-iid settings without mentioning RL.

---

> ### Author Response · Authors · 2020-11-13
> **We added clarifications of the use case importance and the differences from model-based works**
>
> We thank the reviewer for the comments. Indeed, certain clarifications were missing in the text.
>
> By calling our approach "model-free", we mean that we refer to the world as an episodic process that generates rewards with certain joint-distribution - with no further assumptions (certain part of the analysis also assumes normality of that distribution).
> In particular, we avoid 3 assumptions that are highly problematic in real-world applications: (1) fully-observable states; (2) Markov (memoryless transitions); (3) known transitions mechanism. This is now clarified in Section 1.
> Banerjee et al. is indeed an excellent example to the additional value that can be achieved if these assumptions do hold, and we now discuss the difference in Section 7.
>
> Regarding performance improvement, we insist that the current goal of the paper is entirely worthy. Many real-world applications require monitoring without intervention. If an FDA-approved insulin-injecting device fails to balance glucose levels, you don't explore alternative policies - you find out as soon as possible and hurries to the doctor(/engineer). Furthermore, it is a common practice to have a separation between control & safety modules, so you may not even have access to the acting policy. In addition, most ways of intervention in the policy would require further assumptions regarding the process and the policy, which would reduce the scope of validity of the method. This is discussed in Section 1.
>
> We would like to know if the reviewer's opinion could change following the clarification of these two points.
> We also clarified the issues in setup 2 (not assumed to be optimal) and figure 2 (fixed policy).

---

### Decision · Program_Chairs · 2021-01-07
**Final Decision**

**Decision:**

Reject

**Comment:**

The paper's initial evaluation was below par, but the author feedback helped clarify several crucial points after which two of the reviewers increased their scores by a point, bringing the current evaluation to borderline.

The paper addresses a relevant and challenging problem in the RL domain. However, in my opinion, from the reviewers' and authors' remarks and from my own reading of the paper, there are concerns that need to be addressed before the paper can be publication worthy. Primary among these is the quantum of novelty -- as many reviews point out, the key idea of viewing an episodic trajectory as a multivariate (vector) sample for running hypothesis tests is not novel in itself, as is the claim that new tests have been devised. Another crucial issue is the (parametric) assumption of normality for the episodic reward sequence which is not adequately justified in the paper -- even a two time-step trajectory with normal rewards per state transition can exhibit a mixture-of-Gaussians type reward distribution for the second state, breaking the assumption. As it transpired from the reviews of Reviewer4, reducing environment shift/degradation to just a mean change problem, without even considering a change in the variances (2nd order statistics), seems to be too stylized to be effective. There are other, nonparametric approaches in statistics based on testing for changes in the distribution function (kernel density estimation approaches, for instance), which could perhaps be applied without normality assumptions and yield favourable results. The experimental results for detection delay often show significant overlaps of the delay distributions for different procedures (e.g., Hotelling vs. Mean vs. UDT etc.), which does not indicate an advantage of the proposed method.

I would urge the author(s) to assimilate the feedback and delve deeper as to why and how parametric procedures based on normality assumptions may or may not succeed, so as to significantly strengthen the theoretical and practical message of this work.

---

> ### Author Response · Authors · 2021-01-21
> **We believe decision's arguments do not match the reviews**
>
> We thank you for considering our work. However, we disagree with almost all of the reasons that accompanied the decision, and given the open review nature of ICLR, we felt we need to respond to the decision which we further do not feel is supported by the reviews.
>
> 1. A technical note first - "two of the reviewers increased their scores by a point" - in fact 3 reviewers increased their scores, one of them by 2 points.
>
> 2. "is not novel" - while the novelty and relation to the literature were not clearly explained in the original submission, this issue was thoroughly addressed during the rebuttal phase.
> After the revision, the relevant references to this issue were "The proofs seem to be based on standard techniques" (Reviewer1); "significantly improved... including discussing some relevant work" (R3); and "Thanks to the authors for the detailed response and extended discussion on theoretical results" (R4).
> Furthermore, we did demonstrate that the naive mean test is the one that is actually used in practice (see for example the sequential tests paragraph in related work section), and not any multivariate-based test.
>
> 3. "assumption of normality" - we actually do prove that our test is more powerful WITHOUT normality assumption (Theorem 4.2). None of the reviewers raised this issue. In fact, Reviewer4 positively pointed out the very general settings, "such as considering non-Gaussian distributions".
>
> 4. "results... not indicate an advantage" - honestly, we find it difficult to look at the results and think that the different methods are comparable. MDT (which we point out is the most robust version of our test) is considerably better than Mean in 10(!) out of 11 scenarios (where Mean either entirely fails to detect OR has very major detection delay), and both yield similar results in the 11'th scenario. Hotelling also loses to MDT significantly in 8 out of 11 scenarios, and yields similar results in the other 3 scenarios (note that H0 scenario is not to be counted).
>
> This is all shown clearly in Figure 4. The display might be somewhat compressed and we would consider changing it to something cleaner, but the issue of "non-convincing results" has never been raised during the whole reviewing process.
>
> 5. "mean change problem... seems to be too stylized to be effective" - the focus of the test is a fair point for debating (though we wouldn't call a mean-test for degradation "stylized"). However, considering the empirical results, we find it hard to refer to the suggested test as "ineffective”.
>
> Sincerely,
> The authors